# Functional Features of the Respiratory Syncytial Virus G Protein

**DOI:** 10.3390/v13071214

**Published:** 2021-07-01

**Authors:** Larry J. Anderson, Samadhan J. Jadhao, Clinton R. Paden, Suxiang Tong

**Affiliations:** 1Department of Pediatrics, Emory University School of Medicine and Children’s Healthcare of Atlanta, Atlanta, GA 30322, USA; samadhan.jadhao@emory.edu; 2Division of Viral Diseases, National Center for Immunization and Respiratory Diseases, Centers for Disease Control and Prevention, Atlanta, GA 30322, USA; fep2@cdc.gov (C.R.P.); sot1@cdc.gov (S.T.)

**Keywords:** respiratory syncytial virus, G protein, pathogenesis, vaccines, treatment

## Abstract

Respiratory syncytial virus (RSV) is a major cause of serious lower respiratory tract infections in children < 5 years of age worldwide and repeated infections throughout life leading to serious disease in the elderly and persons with compromised immune, cardiac, and pulmonary systems. The disease burden has made it a high priority for vaccine and antiviral drug development but without success except for immune prophylaxis for certain young infants. Two RSV proteins are associated with protection, F and G, and F is most often pursued for vaccine and antiviral drug development. Several features of the G protein suggest it could also be an important to vaccine or antiviral drug target design. We review features of G that effect biology of infection, the host immune response, and disease associated with infection. Though it is not clear how to fit these together into an integrated picture, it is clear that G mediates cell surface binding and facilitates cellular infection, modulates host responses that affect both immunity and disease, and its CX3C aa motif contributes to many of these effects. These features of G and the ability to block the effects with antibody, suggest G has substantial potential in vaccine and antiviral drug design.

## 1. Background

Respiratory syncytial virus (RSV) is a major cause of serious lower respiratory tract infections in children < 5 years of age worldwide, causing an estimated 3.4 million hospitalizations, 95,000–150,000 deaths globally, and up to 175,000 hospitalizations in the United States [1,2]. Nearly all children are infected by 2 years of age, but RSV repeatedly infects and causes disease throughout life with the infants, elderly, and persons with compromised immune, cardiac, and pulmonary systems at the greatest risk of serious disease [3,4]. Given its disease burden, RSV has been a priority for vaccine and anti-viral development for over 50 years. Though no vaccine has yet been developed, immune prophylaxis initially with high RSV antibody titer immune globulin and later a neutralizing monoclonal antibody, palivizumab, is effective and available for young children at high risk for serious RSV disease [5,6].

RSV vaccines are being developed for children < 6 months of age, children of 6 months to 24 months of age, pregnant women, and elderly adults > 65 years of age [7,8]. Some of the obstacles to vaccine development include concern that a non-live virus vaccine for young children may predispose to enhanced RSV disease (ERD); an incomplete understanding of protective immunity and disease pathogenesis; immature or altered immune responses in two target populations, young children, and elderly adults; and the cost of clinical vaccine trials. The first RSV vaccine trial with a formalin-inactivated RSV with alum adjuvant (FI-RSV) led to the concern for enhanced RSV disease (ERD). In this trial, young children who received the vaccine had an unexpected high rate of hospitalization with later RSV infection, and two infected children died [9,10,11,12]. This raised concern that other non-live virus vaccines might also cause ERD and has limited vaccine development for young children to live virus vaccines which are not associated with ERD. In contrast, for adults, ERD is not an issue and, since live attenuated RSV replicates poorly and is not very immunogenic in adults, adult vaccine development has focused on subunit vaccines. Thus, there are two distinct vaccine development strategies, live virus vaccines for children and subunit vaccines for adults.

The past failures suggest that novel approaches to vaccine development may be needed to achieve a successful vaccine [7]. The biology of infection, immunopathogenesis of disease, and protective immunity provide the foundation for developing novel approaches to RSV vaccines. RSV has 10 genes that encode for 11 proteins [13] and two of these proteins, F and G, are associated with inducing protective immunity in animals and neutralizing antibodies [14]. In this review, we will focus on one of these proteins, the G protein, and its role in infection and disease pathogenesis and how these roles inform its potential value to a vaccine and as a target for anti-viral drugs. Of note, there are two major antigenic RSV subgroups, A and B, and substantial strain variation within the two subgroups [15,16,17,18]. Since the F protein is much less variable than G, it induces better cross protection between the two subgroups [19], and, as it induces a higher titer of neutralizing antibody, F has been the focus of most vaccine development. The fact that prior infection and high titers of neutralizing antibodies, including those that are maternally derived, provide some protection from disease, and the effectiveness of immune prophylaxis with a neutralizing F protein monoclonal supports the F protein as important to a vaccine [20,21,22,23,24,25,26].

There are, however, features of the G protein that suggest it could also be an important component of an RSV vaccine [27]. First, from studies in animals, G modulates the host response that contributes to inflammation and disease, and binding G with the antibody prevents much of this disease [28,29,30,31]. Second, it mediates binding to CX3CR1 in primary human airway epithelial cells in vitro as the first step in the infection of cells, and the anti-G antibody effectively neutralizes RSV in these cells [32,33]. Thus, binding G with vaccine-induced or passively administered antibodies or anti-viral drugs has the potential to decrease disease by decreasing virus-induced inflammation and virus replication. Of note, studies in mice with the intact (neutralizes virus) and F(ab’)2 (does not neutralize the virus) forms of an anti-G mAb show that they decrease disease after infection to a similar extent and indicate that a significant portion of the anti-inflammatory effect of binding G does not rely on decreasing virus replication [28,30]. In this review, we provide examples of functional features of G that contribute to these effects and note how this supports G’s potential contribution to an RSV vaccine and as a target for anti-viral drug development.

## 2. Structure of G

The G protein is a type II, highly glycosylated membrane protein of 292–319 amino acids (aa) with an intracellular cytoplasmic tail (aa 1–37), transmembrane domain (aa 38–66), and the extracellular domain ending at its carboxy-terminas (Figure 1).

A second translation initiative site at aa 48 in the transmembrane domain leads to a truncated G that, after the remaining 18 aa in the transmembrane domain are proteolytically cleaved, only contains the extracellular domain [34]. This truncated form of G is secreted. The extracellular domain contains an initial highly glycosylated, a variable mucin-like domain (~aa 66–160), a central relatively conserved domain (aa 160–200), and a second highly glycosylated variable mucin-like domain ending in the carboxy-terminas (ranging from aa 192 to 319). There are 4–5 N-linked glycans and 30–40 O-linked glycans in the extracellular domain of G [34,35]. There are, however, over 75 serine and threonine potential O-linked glycosylation sites and O-linked glycosylation sites usage varies. The G protein is ~32 kDa without post-translational processing, ~95 kDa when fully glycosylated in Hep-2 cells, ~55 kDa cells after glycosylation and cleavage in Vero cells, and ~170 kDa after post-translational processing in primary human bronchial epithelial cells [36]. A virus grown in primary human bronchial epithelial cells is less infectious in Hep-2 cells than a virus grown in Hep-2 cells. The central relatively conserved domain is relatively conserved within but not between the two groups, as illustrated in Figure 1 [27]. For the purpose of this review, we used CCD to indicate this central relatively conserved domain at aa 160–200 of G.

The amino acids 160–200 designated as CCD for this review contain the 13 aa (aa 164–176) that are conserved among all isolates, a CX3C chemokine motif (aa 182–186) [39], and a heparin binding domain (HBD) at aa 187–198 [40]. Through the CX3C motif, G binds to the CX3C chemokine receptor, CX3CR1. One group solved the crystal structure of a subgroup A G peptide from aa 161–197 bound to one monoclonal antibody and the Subgroup A G peptide from aa 169–198 bound to a second mAb [41]. Another group solved the crystal structure of a Subgroup A G peptide from aa 153–197 bound to two different monoclonal antibodies, noted that the structure of this peptide is distinct from fractalkine, and concluded that G likely binds to CX3CR1 differently than fractalkine [42]. Recently, it has been shown that the structure of CCDs is flexible, and this flexibility may allow G to bind to CX3R1 in a fashion sufficiently similar to fractalkine to account for its reported ability to compete with fractalkine binding to CX3CR1 and mimic the fractalkine induction of leukocyte migration in vitro [39,43]. G’s binding to CX3CR1 facilitates the infection of primary human airway epithelial cells and suggests that CX3CR1 is a receptor for the infection of these cells [32,33]. Through the HBD on G, RSV binds to glycosaminoglycans (GAGs) on the cell surface [40]. There may be other HBDs in G, and G HBDs may vary among strains [44,45]. There are also HBDs on F which presumably provide the means for viruses without the G protein to bind to and infect cells [46,47]. F also binds to nucleolin, and nucleolin has been reported to be a receptor for RSV infection [48].

The G protein is the most variable of the RSV proteins, and its antigenic and sequence differences have been used to determine the subgroup and genotype of isolates [49]. Antigenic or sequence differences among RSV strains have made it possible to define many features of the epidemiology and transmission of RSV. For example, early studies with monoclonal antibodies showed that some suspected nosocomial transmissions were nosocomial, while others represented multiple introductions from the community [50,51]. Sequence studies replaced mAb reactivity to identify strain differences and found both in subsequent studies of RSV transmission in healthcare settings [52,53]. G gene sequence studies have also helped understand community RSV outbreaks and shown the co-circulation of multiple genotypes in one season in a community, different patterns of genotypes in nearby communities during the same season, and clarified household transmission [17,18,54,55,56]. One conclusion from these studies is that yearly community RSV outbreak strains are not dominated by regional or national strains. Whole RSV genome sequence studies are expanding our understanding of the fine details of transmission, as indicated in a study of household transmission [57]. Sequence studies of RSV isolates over time in an immune-suppressed child with persistent infection that showed increases in G gene diversity after immune reconstitution associated with the engraphtment of a bone marrow transplant support immune selection contributing to G diversity [58].

Multiple sequence studies of the G gene of isolates from community outbreaks have found specific genotypes associated with increased disease severity [59], but no genotype has consistently been associated with increased disease severity across outbreaks [60]. Thus, G genotypes do not explain strain differences in virulence. Studies in mice show strain-specific differences in virulence associated with sequences in the F protein gene [61,62]. In a recent study, investigators from the Netherlands looked at G gene sequences and their data suggest that specific amino acid changes are associated with increased disease severity [63]. Such links between G gene sequences and disease severity, however, need to be confirmed. It is likely that whole genome sequence studies with genomewide association analysis will be the best way to identify virulence-associated sites in the RSV genome.

Interestingly, a duplication in the G gene of Subgroup B strains was detected in isolates from 1999 in Argentina (RSV BA) [64]. In this strain, aa 240–259 are duplicated and inserted between aa 259 and 260, which results in a 20 aa longer G protein. A G gene duplication in Subgroup A strains was detected in isolates from 2011 in Canada (RSV ON1) [65]. In this strain, aa 261–283 are duplicated and inserted between aa 284 and 285, which results in a 23 aa longer G protein. Viruses with the G duplication, both Subgroup A and B, have spread globally, and variants of the original viruses have become the dominant currently circulating strains [55,66,67]. In vitro studies on reverse genetics-derived Subgroup B viruses with and without the G duplication showed that, the duplication likely improved virus attachment in binding to heparin sulfate proteoglycans on the cell surface and conferred a competitive replication advantage [68]. A study of the in vitro infection of lentivirus pseudoparticles expressing G protein showed that the Group A duplication also improved infectivity [69]. These studies suggest that the selective advantage of the G duplication in Groups A and B may be increased binding to the cell surface, possibly to GAGs, resulting in improved infectivity.

## 3. Secreted G

Secreted G contains all domains of extracellular G and, thus, can interact with cells in a fashion similar to the intact G protein, such as binding to HBD [70], though the effect may be distinct. For example, one group found that RSV with G lacking the secreted G was more pathogenic in mice than the virus with an intact G that had secreted G [71], while another group found that RSV without secreted G was less pathogenic than RSV with intact G in mice [72]. Secreted G has also been reported to enhance cytotoxic T-cell responses in mice [73], act as an antigen decoy, and modulate Fc-mediated antibody anti-viral activity [59,74,75]. Several studies have demonstrated a reduction in proinflammatory responses associated with secreted G but not membrane-bound G. For example, RSV infection of the human airway epithelial cell line, A549, without secreted G compared to RSV with secreted G, showed that the presence of secreted G reduced levels of surface ICAM-1 and secreted IL-8 and RANTES [76]. In another study, the cysteine-rich region within the CCD of secreted G was shown to inhibit F protein and RSV-induced secretion of IL-6 in human peripheral blood monocytes [77]. In this study, they also showed that secreted G appeared to inhibit the early levels of infection-induced IL-6 in lung macrophages as well as lung inflammation in infected mice, and a peptide containing the cysteine-rich region of CCD inhibited the human monocyte response to endotoxin in vitro. Secreted G was also reported to suppress human peripheral blood mononuclear cell lymphoproliferative responses to tetanus toxoid and mycobacterial lysates [78]. These studies highlight the substantial immune modulatory activity of secreted G.

## 4. Binding to and Infection of Cells

The G protein has been noted to bind to a number of molecules on the cell surface, including CX3CR1, through its CX3C motif (aa 182–186) [39], GAGs through its HBDs (including one at aa 187–198 [40]), surfactant A [79], annexin II [80], and DC-Sign and L-Sign [81].

G binding to CX3CR1 in primary human airway epithelial cells (pHAECs) facilitates the infection of these cells, and CX3CR1 is considered a receptor for the infection of these cells [32,33,39,82]. CX3CR1 is a G-coupled transmembrane protein that serves as the receptor for one CX3C chemokine fractalkine [83,84]. CX3CR1 is found on the surface of a number of cell types, such as neurons and microglia cells, smooth muscle cells, airway epithelial cells, monocytes, dendritic cells, NK cells, and T and B cells [85,86,87,88,89]. The ligand for CX3CR1, fractalkine (CX3CL1), has some features similar to the RSV G protein [90,91], i.e., it has a membrane anchored and soluble form and a large mucin-like domain. The soluble form is produced by metalloproteinase Adam 17 cleavage and associated with the directional migration of immune cells to sites of inflammation. The membrane-bound form is associated with cell adhesion. The CX3CRL1-CX3CR1 interaction often has a proinflammatory effect through the JAK-STAT, Toll-like receptor, MAPK, AKT, NF-κB, or other pathways, but can also have an anti-inflammatory effect depending on the tissue, cell type, and local environment [91,92,93]. The CX3CL1-CX3CR1 interaction has been linked to a number of diseases, including cardiac, lung, neoplastic, neurologic, and rheumatologic disease [90,91,94,95]. CX3CR1 has different isoforms, and isoform differences may be responsible for the range of responses associated with fractalkine-CX3CR1 binding [96].

RSV infection of pHAECs is initiated by binding to CX3CR1 through the CX3C motif in G. Some groups have detected GAGs on the surface of pHAECs and the RSV-neutralizing activity of heparin in pHAECs [32,97], while others have not [33,98]. Mouse and human anti-G monoclonal antibodies that bind in CCD-G, i.e., at or near G’s CX3C motif, as well as anti-CX3CR1 antibodies neutralize the RSV infection of pHAECs [32,82,99,100]. Unlike pHAECs, RSV infects continuous cell lines, such as HEp-2 and Vero cells, through GAGs on the cell surface and not CX3CR1. In these cells, heparin effectively neutralizes infection, while anti-G monoclonal antibodies (mAbs) have limited neutralizing activity without the addition of complement [33,40,101]. Note that complement neutralizes the virus through the Fc portion of antibodies bound to the virus and is one of the antibody Fc-mediated anti-viral activities [102] that likely explains anti-G mAb virus neutralization in mice, which requires an intact Fc [28,103,104].

GAGs, e.g., heparin sulfate proteoglycans, are another G-associated receptor for infection and, as noted above, are the receptors that HBDs on G bind to the cell surface and mediate infection [39,105]. There may be other HBDs in G, as indicated by some heparin virus neutralization that is independent of the aa 187–198 HBD and strain variability in HBDs [44,45]. There are also HBDs on F, which presumably provide one way for viruses without the G protein to infect cells [46,47]. Note that the F protein is reported to bind nucleolin and, thus, nucleolin has been proposed as another receptor for RSV infection [48,106].

G is reported to interact with surfactant A [79], annexin II [80], and DC-Sign and L-Sign [81]. This interaction does not mediate infection but does have other effects. For example, surfactant A, as well as surfactant D, have broad antimicrobial effects and can inhibit RSV replication [107,108].

It is also reported that G on the cell surface affects, though is not required for, F-mediated binding to cells, fusion, and possibly other aspects of the role of F in the infection of cells [109,110].

## 5. Animal Studies and Disease Pathogenesis

G affects the host immune response and disease pathogenesis with studies in animals tending to inform G’s effect on disease pathogenesis and in vitro studies G’s effect on the host immune response. The effect on immune responses and disease pathogenesis are interrelated. In this section, we discuss animal model studies that show a substantial role of G in disease pathogenesis. For example, studies with anti-G monoclonal antibodies, anti-CX3CR1 antibodies, and G mutant viruses in mice suggest that G plays a substantial role in the pathogenesis of FI-RSV ERD [111,112]. In the study of FI-RSV vaccinated mice challenged with RSV without G or with G’s CX3C motif mutated to not bind, CX3CR1 showed a marked decrease in pulmonary inflammation and eosinophilia compared to wildtype virus. In this study, they also gave anti-CX3CR1 antibody before challenge with wildtype virus and saw a similar decrease in disease compared to untreated mice. In the second study, the wildtype and F(ab’)2 forms of an anti-G mAb were administered to FI-RSV-vaccinated mice before challenge with wildtype virus, and both decreased pulmonary inflammation and eosinophilia to similar degrees. These studies indicate a role for G at the time of later challenge in FI-RSV-associated ERD. Though it is possible that differences in virus replication accounted for some differences seen with G-altered challenge viruses, the fact that the F(ab’)2 form of the anti-G monoclonal antibody similarly decreased ERD in the mice supports a G effect independent of a decrease in virus replication. The F(ab’)2 form of this monoclonal antibody does not decrease virus replication in the mouse [28,30]. The fact that the anti-CX3CR1 antibody also decreases disease suggests that G binding to CX3CR1 likely explains at least some of G’s role in ERD. Another group reported that anti-G mAbs binding to CCD given before challenge of vaccinia G-vaccinated mice reduced the enhanced disease seen with this vaccine [113]. In a number of studies, the G protein or G peptide vaccines have not predisposed to enhanced disease with later RSV infection in mice [114,115,116,117,118,119]. In other studies of G vaccines in vaccinia virus- or hepatitis B virus-like particles, increased pulmonary eosinophilia was noted with later RSV infection [120,121]. Specific amino acid sequences in G have been associated with pulmonary eosinophilia in mice. In one study, mice vaccinated with vaccinia G and challenged with RSV with a deletion of aa 151 to 221 or aa 178 to 219 compared to challenge with the parent wildtype virus showed a marked in decrease pulmonary eosinophilia [122]. In this study, the viruses with the deletions had similar levels of lung virus post infection as the comparison parent virus. In another study, vaccination with different G peptides with overlapping aa sequences expressed in vaccinia identified aa 193–205 as predisposing to pulmonary eosinophilia with later RSV challenge [123]. These studies indicate that G can sometimes prime for increased disease. On the other hand, G does not need to be present in a FI-RSV vaccine for it to predispose mice to ERD and pulmonary eosinophilia with later RSV challenge [124]. Finally, a CCD peptide vaccine given the day after FI-RSV decreased the ERD and pulmonary eosinophilia with later RSV challenge [125]. The fact that the administration of anti-G antibodies or a G-peptide vaccine can prevent FI-RSV ERD and G is not needed in FI-RSV for it to predispose to ERD suggests that G has a role in FI-RSV ERD at the time of later challenge and not at the time of vaccination.

Studies also show that G makes a substantial contribution to RSV disease during primary infection in mice. For example, administration of the anti-G mAb 131-2G or mAbs with similar specificity, before infection (prophylaxis), reduces weight loss, lung inflammation, and/or lung Th2 cytokine levels [30,126,127]. Additionally, if the infecting virus induces lung mucous production and increased breathing effort (an indicator of airway resistance), mAb 131-2G or a similar mAb given prophylactically also decreases or eliminates these disease markers. Importantly, the F(ab’)2 form of the mAb does not decrease lung virus titer but does effectively reduce these disease markers similarly to the intact mAb [28,30]. Thus, the effect of 131-2G prophylaxis on disease in mice does not require a decrease in virus replication and mAb 131-2G, and likely, antibodies with a similar specificity have two independent effects on RSV disease in mice, an Fc-dependent anti-viral effect and an anti-inflammatory effect. Importantly, the treatment of mice infected with RSV 3 days earlier with mAb 131-2G or a similar mAb promptly decreased weight loss, lung inflammatory cells, lung mucus levels, and/or breathing effort when compared to untreated mice [31,126]. This decrease in lung disease was much quicker and more effective than treatment with palivizumab (the neutralizing anti-F mAb that is licensed for RSV immune prophylaxis in high-risk infants [128]) or a palivizumab-like anti-F mAb. It is likely that the anti-inflammatory effect of these G mAbs was responsible for the prompt decrease in disease, and this G mAb anti-inflammatory effect could be the missing piece in achieving effective RSV treatment. One study suggested that a combination of two anti-G mAbs might be more effective than one. In this study, mAbs reacting at different epitopes in CCD were more effective than alone in decreasing pulmonary inflammation, while an anti-G mAb binding outside CCD was not [129]. Infecting mice with a virus with a mutated G that does not bind to CX3CR1 causes much lower levels of lung disease, which suggests that the G-CX3CR1 interaction is important to G-associated lung disease [130]. In this study, the G-mutated virus replicated to slightly lower levels in the lungs than the wildtype virus but, based on the results with different virus inocula, the decrease in disease was not explained by this difference in lung virus replication.

Interestingly, two groups reported studies of RSV in CX3CR1-deficient mice (CX3CR1-) compared to wildtype CX3CR1+ mice [131,132]. Both studies reported a decrease in the in vitro migration of leukocytes from the CX3CR1- mice. In one study with 6–8 week old mice, there was a decrease in NK cells, neutrophils, and IFN-γ levels in RSV-infected CX3CR1- compared to the CX3CR1+ mice but no difference in lung virus titer [132]. In the other study of neonatal mice, there was an increase in neutrophils, mucus, and IL-17+ γδ T cells in the lungs of RSV-infected CX3CR1- mice and no difference in lung virus titer [131]. These mouse studies illustrate the importance of the CX3CR1-CX3CL1 interaction in immune cell trafficking and support the concept that G’s interaction with CX3CR1 might impact disease.

One feature of RSV infection in infants is apnea, a temporary cessation of breathing [133]. A study of G administered to mice suggested that G might be responsible for apnea, possibly by binding to CX3CR1 and the induction of substance P [134]. In this study, the intravenous administration of G but not F decreased the respiratory rate in mice, while the co-administration of mAb 131-2G, an anti-CX3CR1 mAb, or an anti-substance P mAb, but not an anti-F mAb with G, prevented this decrease in respiratory rate. In other studies, both RSV infection and the G protein have been shown to increase levels of substance P, and this induction of substance P was associated with disease in mice [111,135,136,137].

## 6. In Vitro Studies of the Immune Response

G has been shown to affect a variety of specific immune responses in human cells, which shows G’s potential to affect the human immune response and disease and suggests that at least some of G’s effects seen in animals will apply to human disease. It is not yet clear, however, how these individual effects fit together to explain G’s effect on immunity or disease in humans.

### 6.1. Innate Response

One area of considerable interest is RSV’s effect on dendritic cell responses, including responses that direct the adaptive T-cell response [138], and G appears to have a substantial effect on dendritic cell responses. For example, G binds to DC-Sign and L-Sign on primary human myeloid DCs and plasmacytoid dendritic cells, and this binding induces the phosphorylation of ERK1 and ERK2, which participate in the regulation of dendritic cell responses to infection [81]. In a study of human peripheral blood mononuclear cells exposed to RSV-infected A549 cells, the CX3C motif in G was implicated in the downregulation of the type I and III interferon responses in monocytes and plasmacytoid dendritic cells and interferon gamma in T cells [139]. In an in vitro model of the human response to RSV antigens, the Th1 directing dendritic cell response to RSV or G was increased when the CX3C motif was mutated to CX4C in the virus or in G expressed in VLPs, suggesting that this motif participates in inducing a more Th2-biased immune response, as reported in studies in mice [130,140]. The G protein has also been shown to inhibit the TLR3/TLR4 induction of INF-β in monocyte-derived dendritic cells [141]. In studies of different RSV strains infecting A549 cells and monocyte-derived macrophages, there were differences in IL-6 and CCL5 levels that mapped to the RSV G protein or M2-1 genes [142].

### 6.2. Adaptive Response

Other cells involved in the immune and inflammatory response also are affected by G. For example, viruses lacking G or its CX3C motif were found to increase the trafficking of CX3CR1+ CD4 and CD8 T cells to the lungs of infected mice, suggesting that G and its CX3C motif inhibit the trafficking of these cells to the lungs during RSV infection [143]. Studies with viruses lacking G, secreted G, or the cysteine residues in CCD indicate that G enhances cytotoxic T-cell responses to infection, and this effect is associated with the 4 cysteines in CCD [73,144]. In a study in mice, immunization with a peptide from G’s CCD induced a Th2-biased response, but this peptide fused with an F and M2-1 peptide enhanced the Th1 response, giving a more balanced response than the F or M2-1 peptide alone [145], consistent with G enhancing CTL responses to other antigens and affecting adaptive immunity. More recently, G has also been shown to play a central role in RSV’s modulation of neonatal regulatory B cells (nBreg) and alterations to other responses associated with this effect on nBregs [146]. In this study, RSV F binding to immune globulin on nBregs induced CX3CR1 expression, which then facilitated G RSV infection of nBreg through the G-CX3CR1 interaction. Infection by RSV resulted in IL-10 production which, in turn, altered innate and adaptive immune responses. In a small study of RSV-infected infants, they found RSV-infected nBregs in nasal pharyngeal swab specimens and a positive relationship between the percent of nBregs in blood and disease severity.

### 6.3. Airway Epithelial Cells

G also modulates the response of airway epithelial cells to RSV infection. A study of a G-deleted RSV in a mouse bronchial epithelial cell line indicated that G downregulates suppressor of cytokine signaling 3 (SOCS3), IFN-β, and IFN-stimulated gene (ISG-15) mRNA [147]. In Calu-3 cells, a human airway epithelial cell line, RSV with an inactive CX3C motif induced lower levels of miRNAs, let-7f, and miR-24, and an increase in IFN-λ mRNA [148], suggesting that an intact CX3C motif increases these miRNAs and downregulates INF-λ. Studies of primary human airway epithelial cells exposed to purified G or RSV lacking G suggested that G induces IL-1a and RANTES and inhibits the induction of other cytokines and chemokines, including IP10 and MCP-1 [149]. Another study of the RSV infection of pHAECS cells suggested that the G-CX3CR1 interaction induced RANTES, IL-8, and fractalkine production and downregulated IL-15, IL1-RA, and monocyte chemotactic protein-1 production [32]. These conclusions followed from differences in responses to infection with a virus with an intact or inactivated CX3C motif or the co-administration of anti-CX3CR1 antibodies. In another study of RSV infection of primary airway epithelial cells, significant changes were seen in over 700 transcripts compared to mock infection, including increased nucleolin and decreased cilia-related gene transcripts [150]. The addition of purified G protein to the cells also increased nucleolin and decreased cilia-related gene transcripts. Infection with RSV with an inactivated CX3C motif compared to one with an intact CX3C motif showed a smaller increase in nucleolin and lower decrease in cilia-related gene transcripts, suggesting that the G-CX3CR1 interaction played a role in nucleolin induction and the downregulation of cilia-related genes.

## 7. Conclusions

These studies show the RSV G protein to have wide ranging effects on the biology of RSV infection and the host immune response and disease associated with infection. Though some reports appear contradictory and it is not clear how the in vitro and animal studies noted above translate to human infection nor is it clear how to piece together individual findings to understand RSV immunity, some general conclusions can be drawn. First, G plays an important role in the first step of infection, binding to cell surface molecules through GAGs and/or CX3CR1. Second, G modulates host responses to infection which, in turn, affect immunity and disease. Third, the G-CX3CR1 interaction contributes to both binding to cells and modulating the host response to infection. The fact that binding G with antibodies against CCD decreases lung virus titer, lung inflammation, and disease in mice, and G neutralization is, by a mechanism, distinct from F neutralization, suggests that G can add value to an RSV vaccine. G, or a G peptide, has a potential role in a subunit vaccine, and G’s role in disease pathogenesis suggests that mutations in G might be used to attenuate a live RSV vaccine. The fact that an anti-CCD mAb promptly decreases inflammation and disease during active infection in mice and more rapidly than an anti-F neutralizing mAb does suggests G is a promising target for an anti-viral drug. It is possible that G-target antiviral drugs could succeed when drugs against other targets have failed. Thus, G has substantial potential in both vaccine and antiviral drug design. Further study, however, is needed to determine how G can contribute to the prevention and treatment of human RSV disease.

## Figures and Tables

**Figure 1 viruses-13-01214-f001:**
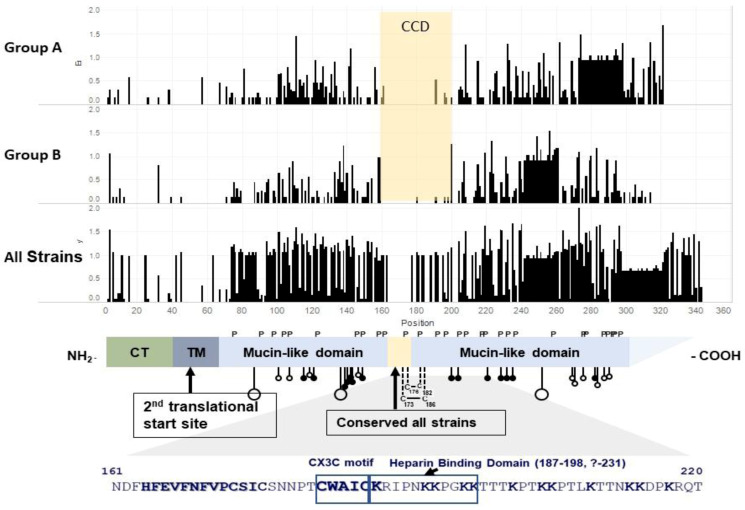
Schematic of the functional domains of the RSV G protein. The first three panels are entropy plots that indicate variability at each amino acid by height of the bar. Sequences were aligned using mafft 7.471, and Shannon entropy was calculated using the formula $H=-\overset{M}{\underset{i=1}{\sum }}P_{i}log_{2}P_{i}$, where *P*_i_ is the fraction of residues of type *I*, and *M* is the total number of residue type (Wootton, J.C. and Federhen, S., *Computers & Chemistry*, 1993) [37], using Python code from https://gist.github.com/jrjhealey/130d4efc6260dd76821edc8a41d45b6a (accessed on 6 April 2021). The panels were generated using Tableau 2021.1.0. The 1st panel, Group A, is based on 50 sequences representing the different Group A genotypes. Since some Group A viruses have a 23 aa duplication in G, a gap was included in viruses without the duplication to maintain the alignment. The 2nd panel is based on 53 sequences representing the different Group B genotypes. Since some Group B viruses have a 20 aa duplication in G, a gap was included in viruses without the duplication to maintain the alignment. The 3rd panel is based on the 103 Group A and B sequences used in panels 1 and 2. It includes one gap for the 23 aa duplication in Group A and a second gap for the 20 aa duplication in Group B viruses. The 4th panel is a schematic of the structure of the RSV A2 G gene adapted from Teng et al. (with permission) [38]. P indicates prolines and C cysteine. The stalk with large circles indicates sites for N-linked carbohydrates, and stalks with small circles indicate sites, serine open, and threonine closed, for O-linked carbohydrates. The 13 aa, aa 164–176, conserved among all strains is highlighted in beige. The 5th panel shows aa sequences that include CCD, the CX3C motif (aa 182–186), and the HBD (aa 187–198). CCD is defined in this review as the central relatively conserved domain from ~aa 160 to 200 that is relatively conserved within but not between Group A and B strains. CT = cytoplasmic tail. TM = transmembrane domain. The CX3C motif and K (lysine) are in bold print.

## Data Availability

The list of sequences used to construct the Shannon entropy in Figure 1 are available upon request to the corresponding author.

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
