# Peer review of "Functional Features of the Respiratory Syncytial Virus G Protein"

_viruses, 2021, doi:10.3390/v13071214_

Round 1

Reviewer 1 Report

Anderson et al. provide a comprehensive review of the G protein, the role it plays in the virus life cycle and modulating the immune response, and as a neutralizing antigen. In several places it would be helpful if the authors would try to make sense of the catalog of experimental results that they list, to provide the reader with a concept of how the virus and antiviral responses are working.

On the pdf, phrases and sentences that seem to be missing something or that are unclear are highlighted.

There is no space between the period and the [ref], except in the first paragraph. What is the standard?

p.1. “…palivizumab” is usually not capitalized.

? “~<6” is odd, please explain.

“Since…live attenuated RSV replicated replicate poorly and [are] not very immunogenic in adults…” does not seem like a good reason to use them in children, as stated. Is it because this is the only option remaining?

p.2. “…binding G with antibody prevents much of this disease.” The addition of antibody to G reduces disease likely by neutralizing G’s function.

Fig. 1. …the first three panels…

“gives” should be “shows”.

p.3. The H and E are both capitalized in “HEp-2”

“Virus grown in primary human bronchial epithelial cells replicates to a much lower titer in Hep-2 cells…” As I understand it, this virus infects HEp-2 cells poorly. In the HEp-2 cells it infects, it replicates fine.

 “Within CCD there is also a chemokine motif, CX3C, at aa182-186, that binds to the corresponding chemokine receptor, CX3CR1[28] and a heparin binding domain (HBD) at aa 187- 198[29].” The CCD includes the first two Cys of the cysteine noose, but not the second two which compose the CX3C motif. Neither is the HBD within the CCD.

“Sequence studies of RSV over time in an immune suppressed child with persistent infection, showed increases in G gene diversity after bone marrow engraftment supporting immune selection.” This sounds like a misinterpretation of the study. Because the child was immunosuppressed there would not be an immune selection. Instead, any mutant that could still replicate would. The deep sequencing would be able to identify most successful viruses. When the child was treated with palivizumab there was a selection for mutations in its binding region.  

p.4. The most prominent GAG on the cell surface is heparan (not heparin) sulfate. It is linked to proteins. The complex is called heparan sulfate proteoglycans.

A large block of text in the last paragraph on this page (highlighted) has a different font. Make sure it is not copied from another article. The same for another long sentence at the top of the next page.

p.5. “…Vero…”

“The need for complement for anti-G antibody to effectively RSV neutralize in cell lines is mirrored by the Fc receptor dependence of anti-G mAb virus neutralization in mice.” The first half of this sentence has no relationship with the second half. The first half describes the requirement for complement to neutralize RSV in cell culture. The second half describes the need for the correct Fc in mice. There are no Fc receptors in Vero cells and Fc receptors are not at all related to complement. The authors need to describe what complement provides in a lab setting that greatly enhances the neutralizing activity of anti-G antibodies which have very little, if any, neutralizing activity in vitro, and what that tells us about the ability of that antibody to directly prevent G’s function. Any antibody bound to a virus can neutralize the virus by lysing the virion membrane. That kills the virus infectivity. Why not mention this well described physiological role as a possibility for how anti-G + complement neutralizes virus when the anti-G doesn’t on its own. The ability of an anti-G antibody to neutralize RSV only in the presence of complement means that the antibody binds to G, but does not directly interfere with its function. A further question is what these anti-G’s do in vivo. There, perhaps the Fc is important for enabling phagocytosis and virus killing by degradation.

Again, heparan sulfate proteoglycans.

“It is also reported that G on the cell surface effects F mediated binding to cells, fusion and possibly other aspects of F role in infection of cells[100, 101].” It should be mentioned that the expression of F alone, without G, causes cell-cell fusion. Then add this first sentence about G perhaps modulating F’s function.

It’s not clear what the authors mean when they say, “[F]or example, studies with RSV without G or with G’s CX3C motif mutated CX4C that does not bind CX3CR1, show either virus given to FI-RSV vaccinated mice gives a marked decrease in the pulmonary inflammation and eosinophilia associated with ERD otherwise seen.” Marked decrease compared to what? If the answer is “compared to wild-type virus”, that would make sense because the infection efficiency of both these mutant viruses is very low (no G and G defective for attachment), which the authors do not mention but should. If instead, the answer is “compared to baseline”, I’m not sure how that would happen. Or is there another infection with wt somewhere in this experiment that isn’t mentioned?    

“Amino acid sequences in G have been associated with pulmonary eosinophilia in mice. In one study, deletion of aa151 to aa221 or aa178 to aa219 in the challenge virus markedly decrease pulmonary eosinophilia that was otherwise seen in mice vaccination with vaccinia expressing G[113].” Again, both these deletions remove the Cys noose, the cell attachment site in the G protein, and would greatly reduce its ability to infect as a challenge virus. Without infection, pulmonary eosinophilia, which would have been caused by infection, would not occur.

“In another study, vaccination of with overlapping G constructs expressed in vaccinia, identified aa193-aa205 as responsible for predisposing to pulmonary eosinophilia with later RSV challenge.” What is meant by “overlapping G constructs”? It sounds like sections of the G protein. Or were they deletions? Where they intact enough to be expressed on the cell surface?

“The fact that administration of anti-G antibodies or a G-peptide vaccine can prevent FI-RSV ERD and G is not needed in FI-RSV for it to predispose to ERD suggests that G has a role in FI-RSV ERD and this role is at the time of later challenge and not as a component of FI-RSV.” Both of these experiments would indicate only that RSV infection is required to trigger ERD since in both cases, the challenge RSV would be neutralized by the G antibodies.

p.6. “Studies also show that G makes a substantial contribution to RSV disease in unvaccinated mice. For example, administration of the anti-G mAb 131-2G or mAbs with similar specificity, before challenge (prophylaxis) reduces weight loss, lung inflammation, and/or lung Th2 cytokine levels.” 131-2G neutralizes RSV’s ability to infect airway cells. Why isn’t that the explanation for inhibiting all the things that RSV infection induces? The authors go on to dismiss the results of this experiment but state that an F(ab2’) form of this antibody neutralizes RSV like the intact antibody but does not prevent heavy breathing or excess mucus production. This first half of the paragraph is very convoluted. They could simplify it by just describing the experiment that was informative. Otherwise the point will be lost.

“The fact that challenge virus with a G mutation that doesn’t bind CX3CR1 causes much less of lung disease in mice described above suggests the G-CX3CR1 interaction is important to the G-associated lung disease.” This paragraph is so long and convoluted that it is not clear what “challenge virus” the authors are talking about. Were these mice immunized before? With what? Or are they referring to the initial infection of the mice with RSV (but using “challenge” for some reason)? If challenge virus is missing its Cys noose in its G protein, it can’t bind to its receptor and it can’t infect. Of course, virus that doesn’t infect causes less virus-induced pathology.

p.7. “G enhances cytotoxic T cell responses to infection and this effect is associated with the 4 cysteines in CCD[63, 137].” As mentioned above the 3rd and 4th Cys are not in the CCD.

The final paragraph begins with “Comment.” Is this meant to be a heading? The whole review would be helped by adding section titles to help orient and guide the reader.

Reviewer 2 Report

This review is a comprehensive presentation of current information about the role of the RSV G protein in infection, pathogenesis, and immune responses to infection.  It is well written, for the most part, and is a valuable contribution to the literature surrounding RSV.  There are minor issues that, if addressed, would facilitate reading of the manuscript.

  1. The figure should be clarified.  The legend should identify the two different inverted lollipops in panel 4 (which I presume are glycosylation sites). 

The identification of P should be added with references. 

The bold Ks in the sequence in panel 5 should be identified. 

The legend should also give a brief indication of the origin of the extra sequences in “all strains” rather than requiring the reader to read subsequent text later in the review for an explanation.  The data is clearly more than just a combination of data in panels 1 and 2 as stated in the legend.

  1. The authors should include a discussion of the glycosylation of the G protein, the two types, their potential functional roles, and references for identification of the glycosylation sites of the non-N linked versions.

  1. The section entitled Disease Pathogenesis is very difficult to read and seems just a list of observations without any overriding theme in their presentation. It would help the reader if the authors could generate and add a table listing major G protein activities perhaps grouping them in related activities.

  1. There are several places where referencing is less than ideal. 

Examples:

Page 2:  the first paragraph has no references

Page 3:  the first paragraph needs more complete references

Page 3:  the first sentence of paragraph 3 needs a reference.

  1. Page 4, line 13: there is a word missing after “without”.

  1. Page 5, line 2: is CL3CL1 correct?

  1. Page 6, paragraph 3: monoclonal antibody is identified as an antibody against CX3CR1 but in paragraph 1 it is identified as an anti-G mAb.

Author Response

This review is a comprehensive presentation of current information about the role of the RSV G protein in infection, pathogenesis, and immune responses to infection.  It is well written, for the most part, and is a valuable contribution to the literature surrounding RSV.  There are minor issues that, if addressed, would facilitate reading of the manuscript.

  1. The figure should be clarified.  The legend should identify the two different inverted lollipops in panel 4 (which I presume are glycosylation sites). 

Response. The legend for the legend for panel 4 has been expanded and clarifies the number of figure designations including the inverted lollipops. 

The identification of P should be added with references. 

Response. I assume the P they are referring to is in the figure.  This is now noted and the source noted.   

The bold Ks in the sequence in panel 5 should be identified. 

Response. The bold K is now defined in the legend relating to panel 5.   

The legend should also give a brief indication of the origin of the extra sequences in “all strains” rather than requiring the reader to read subsequent text later in the review for an explanation.  The data is clearly more than just a combination of data in panels 1 and 2 as stated in the legend.

Response. We have clarified the source of extra sequences in the 3 panels in the figure legend.   

  1. The authors should include a discussion of the glycosylation of the G protein, the two types, their potential functional roles, and references for identification of the glycosylation sites of the non-N linked versions.

 Response. We have expanded the discussion of the glycosylation of the G protein as suggested and consistent with the purpose of this review.

  1. The section entitled Disease Pathogenesis is very difficult to read and seems just a list of observations without any overriding theme in their presentation. It would help the reader if the authors could generate and add a table listing major G protein activities perhaps grouping them in related activities.

Response. We have revised the Disease Pathogenesis section and now include topic headings. These changes should guide the reader in reading this section.

There are several places where referencing is less than ideal. 

Examples:

Page 2:  the first paragraph has no references

Response. Appropriate references have been added.   

Page 3:  the first paragraph needs more complete references

Response. We have added references as suggested.   

Page 3:  the first sentence of paragraph 3 needs a reference.

Response. We have added a reference and related the statement to the figure which supports the concept present.

  1. Page 4, line 13: there is a word missing after “without”.

 Response. Corrected (p. 6). 

  1. Page 5, line 2: is CL3CL1 correct?

 Response. Should be CX3CL1.  Corrected. (p. 7)   

  1. Page 6, paragraph 3: monoclonal antibody is identified as an antibody against CX3CR1 but in paragraph 1 it is identified as an anti-G mAb.

Response. We have added a sentence to explain the role of the anti-CX3CR1 antibody in this paragraph. (p. 8)

Round 2

Reviewer 1 Report

Looks good.